# Outcome of Kidney Transplants from Viremic and Non-Viremic Hepatitis C Virus Positive Donors into Negative Recipients: Results of the Spanish Registry

**DOI:** 10.3390/jcm12051773

**Published:** 2023-02-23

**Authors:** Antonio Franco, Francesc Moreso, Eulàlia Solà-Porta, Isabel Beneyto, Núria Esforzado, Francisco Gonzalez-Roncero, Asunción Sancho, Edoardo Melilli, Juan Carlos Ruiz, Cristina Galeano

**Affiliations:** 1Nephrology Department, Hospital General Universitario, 03010 Alicante, Spain; 2Nephrology Department, Hospital Universitario Vall d’Hebron, Department of Medicine, Universitat Autònoma Barcelona, 08035 Barcelona, Spain; 3Nephrology Department, Hospital del Mar-Parc de Salut Mar, 08003 Barcelona, Spain; 4Nephrology Department, Hospital Universitari i Politècnic La Fe, 46026 Valencia, Spain; 5Renal Transplant Unit, Nephrology Department, Hospital Clínic, 08036 Barcelona, Spain; 6Nephrology Department, Hospital Universitario Virgen del Rocío, 41013 Sevilla, Spain; 7Nephrology Department, Hospital Universitari Doctor Peset, 46017 Valencia, Spain; 8Nephrology Department, Hospital Universitari Bellvitge, L’Hospitalet, 08907 Barcelona, Spain; 9Nephrology Department, Hospital Universitario Marqués de Valdecilla, 39008 Santander, Spain; 10Nephrology Department, Hospital Universitario Ramón y Cajal, 28034 Madrid, Spain

**Keywords:** kidney transplantation, hepatitis C virus, viremic donor, graft outcome, hepatocellular carcinoma

## Abstract

Historically, donor infection with hepatitis-C virus (HCV) has been a barrier to kidney transplantation. However, in recent years, it has been reported that HCV positive kidney donors transplanted into HCV negative recipients offer acceptable mid-term results. However, acceptance of HCV donors, especially viremic, has not broadened in the clinical practice. This is an observational, multicenter, retrospective study including kidney transplants from HCV positive donors into negative recipients reported to the Spanish group from 2013 to 2021. Recipients from viremic donors received peri-transplant treatment with direct antiviral agents (DAA) for 8–12 weeks. We included 75 recipients from 44 HCV non-viremic donors and 41 from 25 HCV viremic donors. Primary non function, delayed graft function, acute rejection rate, renal function at the end of follow up, and patient and graft survival were not different between groups. Viral replication was not detected in recipients from non-viremic donors. Recipient treatment with DAA started pre-transplant avoids (*n* = 21) or attenuates (*n* = 5) viral replication but leads to non-different outcomes to post-transplant treatment with DAA (*n* = 15). HCV seroconversion was more frequent in recipients from viremic donors (73% vs. 16%, *p* < 0.001). One recipient of a viremic donor died due to hepatocellular carcinoma at 38 months. Donor HCV viremia seems not to be a risk factor for kidney transplant recipients receiving peri-transplant DAA, but continuous surveillance should be advised.

## 1. Introduction

Hepatitis C virus (HCV) is an RNA virus from the Flaviridae family with seven different genotypes [1]. Parenteral transmission of HCV has been well-documented and renal transplants recipients from HCV positive donors can acquire the infection and develop acute and chronic hepatitis [2]. For this reason, until recently, renal transplantation from HCV positive donors has not been done in non-infected recipients. The report of the American Society of Transplantation consensus conference on the use of HCV positive donors in solid organ transplantation states that, in general, donors with positive HCV antibodies without viremia do not transmit the infection; thus, historical data of HCV “positive” donors must be viewed as limited since it does not specifically differentiate the presence or absence of viremia. In this document, it is recommended to abandon the term HCV “positive” donor and to include the term HCV viremic donor which requires a nuclear acid testing (NAT) result and allows us to differentiate the presence of viremia from its absence, improving the infection transmission risk. Thus, the term HCV viremic donor should be adopted to replace the term HCV positive donor. Additionally, despite this document’s support of the proposition that the policy of kidney transplantation from HCV viremic donors into non-viremic recipients should be conducted under Institution Review Board (IRB)-approved protocols with a rigorous, multi-step informed consent process [3], transplantation from HCV viremic donors into non-viremic recipients was adopted as the standard of care in many centers from US Importantly, different prospective studies have documented the safety and efficacy of the direct-acting antivirals (DAA) drugs to transplant kidneys from HCV viremic donors into negative recipients (reviewed in [4,5]). Finally, the pangenotype efficacy of the combination glecaprevir/pibrentasvir allows the initiation of treatment before HCV genotype identification [6,7].

In 2019, we report the first European experience with HCV viremic/non-viremic donors into HCV negative recipients conducted at three Spanish hospitals [8]. We describe four recipients from HCV viremic donors receiving an 8-week course of the combination glecaprevir/pibrentasvir started pre-operatively, and seven patients receiving kidneys from HCV non-viremic donors which were managed conservatively. Our data suggest that renal transplantation from HCV positive donors into HCV negative recipients is safe when only recipients of organs from HCV viremic donors are treated [9]. This early experience inspired the Spanish consensus document coordinated by the National Transplant Organization (ONT) [10]. In this document, it is realized that between 2011 and 2017, from 246 HCV positive potential donors, only 31% were finally effective donors, and it was estimated that 5–10 HCV viremic donors can be obtained yearly. A specific patient informed consent was provided for this type of donor and universal treatment with DAA was covered by our national health system. For renal transplants recipients, it was suggested to employ DAA with a pangenotype efficacy from the pre-operative period. However, currently, only 24 out of 43 adult renal transplant units in Spain accept kidneys from HCV non-viremic donors and 9 accept HCV viremic donors.

The aim of the present study is to compare clinical outcomes of HCV negative renal transplant recipients receiving organs obtained from HCV viremic/non-viremic donors, reported to the Spanish group.

## 2. Patients and Methods

### 2.1. Patients

This is a retrospective, observational study conducted in 24 renal transplant units from Spain. All donors with a positive HCV serology between 2013 and 2021 were identified and NAT testing was done before organ retrieval (XPERT HCV/HIV load test, Cepheid Inc., Sunnyvale, CA, USA). Donors were identified as HCV viremic or non-viremic and this information was available for the clinicians before transplantation. Donors who were HIV positive, active intravenous drug-abusers, and institutionalized persons during the last year were discarded. Recipients from HCV viremic donors signed a written informed consent form defined by the Spanish consensus document [10] and were treated with a combination of glecaprevir 300 mg/day and pribentasvir 120 mg/day for 8–12 weeks regardless of the presence of recipient viremia. Treatment can be started pre-operatively to complete an 8-week course or until 10 days after surgery once HCV replication was confirmed to complete a 12-week course. HCV viral load was determined approximately at +7, +14, +21, +30, +45, +60, +90, +120, and +180 days. Sustained virologic response (SVR) was defined as a negative NAT result 12 weeks after ending treatment with DAA. For HCV non-viremic donors, NAT monitoring was done at the same intervals and treatment was started in the case of a positive result. Recipient liver enzymes (AST and ALT) were recorded as per local practice at each visit.

Induction and maintenance immunosuppression was guided according to local practices. The following variables were recorded: donor and recipient demographics (age and sex); recipient’s weight; donor/recipient blood group; brain death donor or donor after circulatory death; first transplant vs. re-transplant; last calculated panel reactive antibodies (cPRA) by Luminex assay; ABDR HLA matching; and cold ischemia time.

The present study was performed in accordance with the Declaration of Helsinki and is consistent with the Principles of the Declaration of Istanbul on Organ Trafficking and Transplant Tourism.

### 2.2. Outcome Variables

The following primary outcome variables were recorded: patient and graft survival at the end of follow up as well as HCV viral load 1 week after transplantation and 12 weeks after completing DAA treatment (sustained virologic response: SVR). The following secondary outcome variables were recorded: primary non-function including early vascular thrombosis; delayed graft function defined as dialysis requirement during the first week after transplant once vascular and/or urinary tract complications were ruled out; biopsy-proven acute rejection; estimated glomerular filtration rate according to CKD-EPI formula at the end of follow up. The rate of HCV seropositive recipients at the end follow-up and the evolution of liver enzymes (AST and ALT) after transplant were also analyzed. Finally, adverse events related with DAA and treatment interruptions of DAA were also recorded.

### 2.3. Statistics

Variables were described as frequencies, median and interquartile range, or mean and standard deviation for categorical, non-normally distributed continuous variables and normally distributed continuous variables, respectively. To compare data between groups, Fisher’s exact test, Mann–Whitney U test, Kruskal–Wallis test and Student’s *t*-test were employed according to the variable distribution. Graft and patient survival were analyzed by means of Kaplan–Meier curves and comparison between groups was done by the log-rank test. All analysis were two-tail and a *p*-value < 0.05 was considered significant.

## 3. Results

### 3.1. Patient Characteristics

During the study period, 69 HCV positive donors were used to perform 116 kidney transplants into seronegative recipients in 24 Spanish renal transplant units. A total of 75 kidney transplants were done from 44 non-viremic donors and 41 transplants from 25 viremic donors. From non-viremic donors, 5 kidneys were employed to transplant seropositive recipients while 8 were discarded by surgeons (inadequate perfusion, vascular damage, or macroscopical aspect). From viremic donors, three kidneys were employed to transplant seropositive recipients and six kidneys were discarded by surgeons. This means that the efficacy rate in this pool of donors was 91% for non-viremic donors (80 out of 88) and 88% for viremic donors (44 out 50).

Donor, recipient, and transplant-related variables in kidney transplants from non-viremic and viremic donors were not different (Table 1). In transplant recipients from viremic donors, DAA treatment was started pre-transplant (*n* = 26) or during the initial 10 days after confirming viral transmission (*n* = 15), depending on the renal transplant unit preferences.

### 3.2. Outcome Variables

Patient and graft survival were not different between groups (Figure 1 and Figure 2). In the non-viremic group, 5 patients out of 75 died because of sudden death (2 recipients), acute pancreatitis, sepsis, and COVID-19 infection at 1, 2, 2, 12, and 24 months, respectively. In the viremic group, 4 patients out of 41 died because of sepsis, coronary disease, COVID-19 infection, and hepatocellular carcinoma (HCC) at 1, 12, 14, and 38 months, respectively. Two renal transplants from non-viremic donors failed due to early vascular thrombosis and there was one never functioning kidney. Five renal transplants from non-viremic donors experienced late failure due to recurrent primary disease (*n* = 1), non-treatment compliance (*n* = 1), chronic rejection (*n* = 2), and chronic allograft dysfunction (*n* = 1). Three renal transplants from viremic donors failed due to early vascular thrombosis (*n* = 1), chronic rejection (*n* = 1), and chronic allograft dysfunction (*n* = 1).

Patient and graft survival were also not different in recipients from a viremic donor starting DAA treatment before or after surgery (Figure 2).

Noticeably, there was a 72-year-old male recipient dying with a functioning graft at 38 months due to HCC. He had adult polycystic kidney disease and received a first deceased donor transplant in July 2018 from an expanded criteria donor who experienced early vascular thrombosis. Later, he received a second graft from a brain-dead 50 years-old male viremic donor (5.6 log). The recipient received an 8-week course of glecaprevir/pribentasvir started before transplant. Immunosuppression was based on the combination of anti-thymocyte globulin, tacrolimus, sirolimus, and corticosteroids. Clinical course after transplantation was uneventful and he reached a nadir serum creatinine of 1.3 mg/dL. HCV viral load was negative at all time points and a mild increase of liver enzymes was detected at 2 weeks (AST 56 and ALT 190 UI/L) which normalized thereafter. At 30 months, he was admitted to another hospital due to progressive weight loss, diarrhea, and acute renal failure. Liver ultrasound showed multiple nodular hypoechogenic lesions (8 cm involving segments V-VI, 3 cm in segment V, and 2 nodules of 1 cm in segment III). A liver biopsy of the largest lesion yielded the diagnostic of HCC. Immunohistochemistry was positive for cytokeratin 8/18 and negative for cytokeratin 7 and alpha-fetoprotein. The viral load in the liver biopsy was negative. Hepatitis B virus surface antigen (HBsAg) and antibody to core antigen were negative, and alpha fetoprotein serum levels were ×10 times the upper normal limit. Despite a short treatment with sorafenib, the clinical condition of the patient grew progressively impaired, and he died few months after diagnosis.

Recipients from non-viremic donors showed a negative HCV viral load at all time points since the beginning of transplant. Recipients from a viremic donor starting DAA treatment pre-transplant did no showed viral replication at any time (*n* = 21) or a very low replication (<2 log) at 7 days (*n* = 5). Recipients from a viremic donor starting DAA after transplantation (*n* = 15) showed transmission in all cases with a viral load at approximately 7 days of 5.1 ± 1.0 log. From 14 days after transplant to the end of follow up, all recipients from viremic donors, either starting DAA treatment before or after transplant, had a negative HCV viral load. Thus, the SVR for treated patients was 100%.

Primary non-function, delayed graft function, biopsy-proven acute rejection, and eGFR at the end of follow up were not different between groups (Table 2).

The rate of HCV seroconversion was 100% (14 cases) in recipients from viremic donors starting DAA treatment after transplantation, 56.2% (13 out of 23) in recipients from viremic donors starting DAA treatment before transplantation, and 16.4% (12 out of 73) in patients receiving grafts from HCV non-viremic donors (*p*-value < 0.001). There were 6 cases in whom serology for HCV was not done. Evolution of liver enzymes (AST, ALT) according to the presence of donor viremia and DAA treatment is shown in Figure 3. ALT tended to be higher at 30 days in patients receiving a graft from a viremic donor starting treatment after transplantation, but this difference did not reach statistical significance (*p* = 0.069). Seroconversion rate and evolution of liver enzymes were not different according to induction therapy at the time of transplant.

All patients completed treatment with DAA without reported adverse events and without treatment interruptions.

## 4. Discussion

We report a large experience of HCV seronegative recipients receiving kidneys from HCV positive donors, either non-viremic or viremic. In recipients from non-viremic donors, active surveillance of viral transmission was pursued; in recipients from viremic donors, peri-transplant treatment with DAA was started. Both cohorts were not different in their baseline characteristics and patients and graft outcomes were not significantly different. Noticeably, we report the first case of an HCV seronegative renal transplant recipient receiving a graft from an HCV viremic donor who develops an HCC after transplantation. Despite a link between transplantation with a viremic donor and development of HCC, it cannot be stablished in our case; this observation may recommend continuous liver surveillance of these patients. Undoubtedly, new reports of large series using HCV positive viremic donors will contribute to establish whether this association was pathogenic.

Since the initial reports in the US and Europe, HCV positive donors have been progressively employed as kidney donors in different countries [11,12]. In 2019, we report the first series in Europe using HCV non-viremic and viremic donors, and we conclude that renal transplantation from HCV positive donors into HCV negative recipients is safe when only the recipients of organs from viremic donors are treated [8]. This preliminary experience led the Spanish National Organization (ONT) to establish a protocol to guide patient-informed consent and treatment with DAA for recipients of viremic donors [10]. Despite this protocol, there are a significant number of renal transplant units in Spain not accepting HCV positive donors, especially viremic donors. Thus, this registry will contribute to defining the outcomes of these transplants and to encourage other renal transplant units to employ them. Both cohorts, viremic and non-viremic, are homogenous and like the populations of donors and recipients in Spain. We decided not to include a matched population of HCV seronegative donors and recipients, since national registries are available for comparisons. In fact, patient and graft survival were not significantly different to the last report of the Catalan registry of renal transplants [13]. Patient survival at 1 and 5 years in this registry is 94.9% and 85.1%, while in our cohorts it was 94.6% and 89.5%. Similarly, graft survival including patient’s death in the Catalan registry is 87.1% and 68% at 1 and 5 years, respectively, while in our cohorts these figures were 90.3% and 74.3%. Other outcomes included in our study such as delayed graft function, biopsy-proven acute rejection rates or renal function, were also non-different.

One of the most intriguing observations of our study is that one recipient develops a fatal HCC early (30 months) during follow up. This is the first case reported in the literature of this scenario. In a recent review of HCV-viremic donors into HCV-negative recipients [4], such a serious event was not reported in over 300 kidney transplants. Until now, the pathogenesis of HCV-induced HCC and the HCC risk after DAA cure were incompletely understood. HCV is an RNA virus with little potential for integrating its genetic material into the host genome and it has been proposed that HCV contributes to hepatocarcinogenesis through direct and indirect ways. HCV-mediated liver disease and carcinogenesis are considered multistep processes that include chronic infection-driven hepatic inflammation and progressive liver fibrogenesis with the formation of neoplastic clones that arise and progress in the carcinogenic tissue microenvironment. A gene expression signature in the liver tissue of HCV-infected patients has been associated with HCC risk and mortality, suggesting that virus-induced transcriptional reprogramming in the liver could play a functional role in hepatocarcinogenesis [14]. Moreover, epigenetic changes associated with liver cancer persist after sustained virologic response [15]. From the clinical point of view, it has been widely described that although the viral cure decreases the overall HCC risk in HCV-infected patients, it does not eliminate virus-induced HCC risk, especially in patients with advanced fibrosis [16,17]. In our case, there was no known preexisting liver disease and the patient only presented mild transient elevation of liver enzymes after transplantation, while viral replication in blood was not detected at any time since DAA treatment was started pre-transplant. Unfortunately, once the diagnostic was reached, the disease was too advanced, with multiple large nodules in the liver. Since our patient received chronic immunosuppression, we cannot discard that this fact increases the risk of this serious complication. Despite the patient receiving induction treatment with thymoglobulin, in our study, there were other patients receiving a kidney from a viremic donor treated with thymoglobulin who did not develop liver complications. Similarly, in the studies conducted in the US a significant proportion of patients were also treated with thymoglobulin [18]. Thus, whether surveillance by liver enzymes, serum biomarkers (alpha-fetoprotein), and ultrasound are advisable for these patients should be decided by their treating physicians.

Our study confirms some data already reported. Kidney transplant from HCV non-viremic donors is a safe procedure and does not require treatment since viral transmission is exceptional [19,20]. It is important to emphasize that in our cohort, we exclude active intravenous drug-abusers and institutionalized persons to reduce the risk of transmission of other viruses such as HIV; thus, the probability of being in the window period after HCV transmission is further reduced [21,22]. Kidney transplantation from HCV viremic donors is associated with a very high rate of transmission. Despite the Spanish consent document suggesting starting treatment with DAA before transplant, some centers decided to start treatment during the first week once viral replication was detected. Pre-transplant DAA treatment prevents early viral replication in a lot of cases (21 out of 26) or reduced its intensity to a very low viral load (5 out of 26). Conversely, kidney transplantation from HCV viremic donors into seronegative recipients led to universal transmission without DAA treatment as it has been previously described [23,24,25,26]. In any case, early treatment with DAA is associated with rapid clearance of viral replication and, in our study, it was associated with non-different outcomes to those which have been previously described [24,27].

We observed a high rate of seroconversion in patients experiencing HCV viremia (100%), but also in recipients from HCV viremic donors who never had viremia (56%) and even in recipients from non-viremic donors (16%). This observation has been previously described and the rate of seroconversion of recipients from HCV non-viremic donors under close monitoring may be as high as 44% [20]. In this study the authors concluded that the reason(s) for seroconversion is(/are) unclear, but this phenomenon does not appear to indicate HCV transmission. Later, Porrett et al. conducted a nice study in HCV naïve kidney and heart transplant recipients from HCV viremic donors to address this question. They concluded that the IgG isotype of this antibody (not IgM) and the kinetics of its appearance and durability suggest that the anti-HCV antibody is donor derived and likely produced by a cellular source. They suggest that transfer of donor humoral immunity to a recipient may be much more common than previously appreciated [28].

The evolution of liver enzymes, AST and ALT, was not significantly different among groups despite the trend to higher ALT levels at 30 days in transplants receiving a graft from a viremic donor and starting DAA treatment after transplantation. In kidney transplants, increase of AST/ALT is a frequent finding, affecting up to 24% of patients depending on the series. The main causes during the first months after transplant are attributed to hemodynamic alterations during the surgical procedure, viral infections, previous liver disease, and pharmacological hepatotoxicity (immunosuppressants including thymoglobulin, antibiotics, and antiviral agents). Additionally, the initial kidney injury and the kidney–liver crosstalk may cause hepatic damage and might be responsible for this increase. Recently, it has been described that the increase of AST/ALT is a frequent and transient event related to the kidney donor type, being more frequent in recipients from uncontrolled DCD that normalizes one month after transplant [29]. Thus, we cannot discard that the mild elevation of ALT in transplants receiving a graft from a viremic donor and starting DAA treatment after transplantation reflects hepatocellular damage related to HCV infection. However, we did not observe cases of fibrosing cholestatic hepatitis in recipients from viremic donors who started treatment either before or after transplantation.

Noticeably, in our country, the scenario is different from the US It should be noted that HCV-viremic donors are younger in the US than in our cohort (32 vs. 55 years). Additionally, in the US cohorts, HCV-viremic donors were persons who injected drugs and died due to drug-overdose [30], while these kinds of donors were rejected in our country and in Germany [8,12]. For these reasons, in Europe, the number of non-viremic HCV donors is higher than the number of viremic ones. This different criterion to accept viremic donors is related to the fact that the rate of fatal drug-related overdose is much lower in Europe than in the US (20.6 vs. 2.3 per 100.000 in 2017) [31].

Our study has some limitations related to a multicenter registry, the heterogenicity in the immunosuppression and, especially, in the timing (before or after transplantation) of the initiation of DAA treatment. Unfortunately, histological assessment of pre-implantation kidney biopsies to evaluate organ quality from HCV positive donors was not done, despite new tools to characterize it impending [32]. Additionally, we cannot rule out a selection bias in our cohort, since a contemporary control group receiving a kidney transplantation from HCV negative donors was not included. However, it allows us to confirm non-different outcomes in HCV seronegative renal transplant recipients receiving grafts from viremic or non-viremic donors. Additionally, all patients were treated with the same combination of DAA (glecaprevir/priventasbir), which are pangenotypic and can be safely employed in patients with chronic renal failure [7].

In summary, we reported the analysis of HCV naïve recipients of a renal graft from non-viremic and viremic HCV donors and we described that patient and graft outcomes were not different between groups. Recipients from viremic donors treated with DAA before or early after transplantation also showed non-different outcomes. Despite one recipient of a viremic donor developing an early HCC, we cannot establish a link between both events.

## Figures and Tables

**Figure 1 jcm-12-01773-f001:**
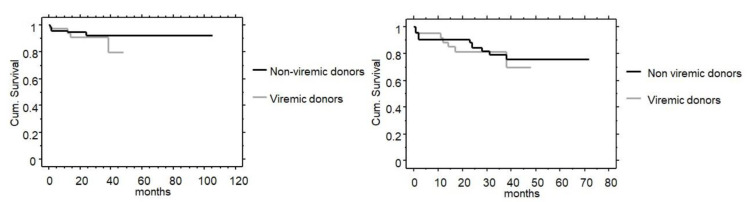
Patient (**left**) and graft survival including patients’ death (**right**) in recipients from non-viremic and viremic donors. *p*-value were non-significant by Kaplan–Meier analysis (*p* = 0.415 and *p* = 0.741, respectively).

**Figure 2 jcm-12-01773-f002:**
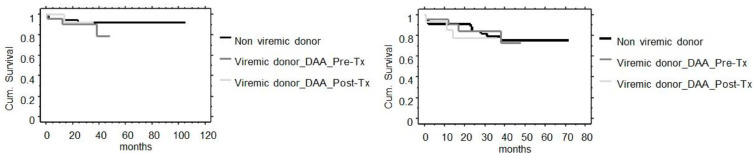
Patient (**left**) and graft survival including patients’ death (**right**) in recipients from non-viremic and viremic donors either starting DAA treatment pre-transplant or post-transplant. *p*-values were non-significant by Kaplan–Meier analysis (*p* = 0.614 and *p* = 0.768, respectively).

**Figure 3 jcm-12-01773-f003:**
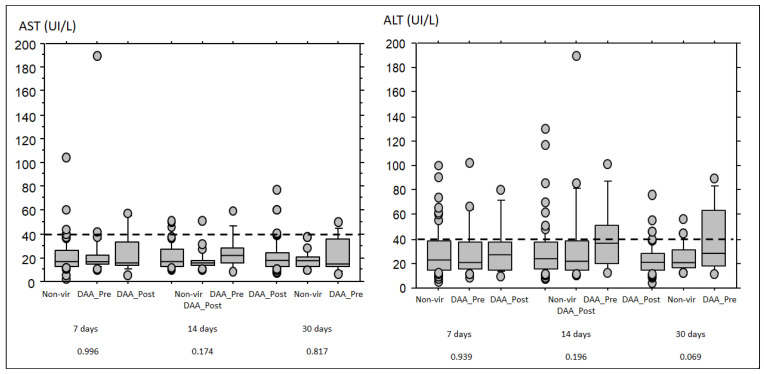
Box plot of the evolution of liver enzymes (AST (**left**) and ALT (**right**)) in patients receiving grafts from non-viremic donors (*n* = 75) and viremic donors either starting DAA treatment before (*n* = 26) or after transplantation (*n* = 15). *p*-values by Kruskal–Wallis test are depicted at the bottom of the figure.

**Table 1 jcm-12-01773-t001:** Donor and recipient demographics and transplant-related variables among transplants performed with hepatitis C virus positive non-viremic and viremic donors.

Variable	Non-Viremic Donors	Viremic Donors	*p*-Value
Donors (*n*)	44	25	
Recipients (*n*)	75	41	
Donor age (years)	56 ± 15	54 ± 7	0.458
Donor gender (m/f)	45/30	33/8	0.038
Donor type (BDD/cDCD)	62/13	38/3	0.169
HCV viral load (log)	n.a.	5.5 ± 1.2	n.a.
Recipient age (years)	57 ± 11	54 ± 10	0.167
Recipient gender (m/f)	53/22	28/13	0.777
Recipient weight (kg)	72 ± 14	76 ± 17	0.253
Primary renal disease (CGN, CTIN, ADPKD, DN, vascular, other, unknown)	16/7/10/11/5/9/17	7/2/6/2/1/10/13	0.166
ABO group (0/A/B/AB)	37/32/5/1	23/14/3/1	0.819
Number of transplant (1/>1)	67/8	33/8	0.259
ABDR HLA donor-recipient mismatches	4.1 ± 1.1	4.4 ± 1.1	0.272
Last cPRA (%)	11 ± 28	11 ± 28	0.861
Cold ischemia time (hours)	17 ± 5	16 ± 6	0.414
Induction therapy (Basiliximab/Thymoglobulin/none)	36/24/16	19/10/12	0.545

BDD, brain death donor; cDCD, controlled donation after circulatory death; HCV, hepatitis C virus; CGN, chronic glomerulonephritis; CTIN, chronic tubule-interstitial nephritis/chronic pyelonephritis; ADPKD, autosomal dominant polycystic kidney disease; DN, diabetic nephropathy; HLA, human leukocyte antigen; cPRA, calculated panel reactive antibodies; n.a., not applicable.

**Table 2 jcm-12-01773-t002:** Outcome variables from transplants performed with hepatitis C virus positive non-viremic and viremic donors.

Variable	Non-Viremic Donors	Viremic Donors	*p*-Value
Recipients (*n*)	75	41	
Primary no function (%)	3 (4.2%)	1 (2.4%)	0.999
DGF (%)	22 (29.3%)	14 (34.1%)	0.745
BPAR (%)	9 (12%)	1 (2.4%)	0.095
Graft loss (%)	13 (17.3%)	7 (17.1%)	0.999
Patient death (%)	5 (6.8%)	4 (9.7%)	0.718
Last eGFR (mL/min/1.73 m^2^)	49 ± 17 (*n* = 62)	46 ± 16 (*n* = 34)	0.354
Follow up (months)	25 ± 16	21 ± 14	0.109

DGF, delayed graft function; BPAR, biopsy-proven acute rejection; eGFR, estimated glomerular filtration rate by CKD-EPI formula.

## Data Availability

The data presented in this study are available on request to the corresponding author.

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
