# Peer review of "Outcome of Kidney Transplants from Viremic and Non-Viremic Hepatitis C Virus Positive Donors into Negative Recipients: Results of the Spanish Registry"

_jcm, 2023, doi:10.3390/jcm12051773_

Round 1

Reviewer 1 Report

This study prospectively collected the data of 116 HCV negative kidney transplant recipients that received their graft from HCV positive donors (44 non viremic donors and 25 verimic donors) in 21 centers.

This study is well written and add a plus-value as compared to other articles on this subject with the adding of NAT+ donors

My concerns :

-       Reasons for discarded kidney organs ? The percentage of discarded kidney graft is more important in NAT+ donors. Was it because of HCV or because of donor characteristics ?

Title of figure 1 : please add “right panel” for graft survival. Is it death-censored graft survival or raw-survival ? Time is mentioned but  “months” should be add to the legend.

-       Biopsy proven rejection include antibody and cellular rejection ?

-       Do the author have an idea of the Donor specific antibody de novo rate ? It should be included in the article.

-       What is the follow-up time for each group ? is it comparable ? The resultas of Table 2 are difficult to interpret without a period of follow-up

-       How was assessed side effects? was it a retrieval in medical files or a standardized document assessed at defined times ?

-       What were the indication of Basiliximab versus thymoglobulin induction ? do the author can assess the results of seroconversion / elevated liver enzyme and HCV viremia according top induction ? i.e. is there an over risk with thymoglobulin ?

-       I agree with author in the discussion that lover enzymes are not specific enough to assess the liver damages. Another test would have been interesting and more relevant (elastography ? CT-Scan ?)

-       I am not sure we can say that it is a prospective study because the protocol are not the same in all centers

Reviewer 2 Report

The paper is focused on a relevant topic. The manuscript is clear and presented in a well structured manner. Materials and methods are described in detail. Results are reported clearly and appropriate. Tables and figures properly show the data. The discussion is adequate with current citations. The conclusions are consistent with the evidence. 
Can authors add infomations regarding the transplanted kidney and in specific regarding if any histological assessement had been assessed? This is a great, debated topic that can have great impact on outocomes. Expertise, type of specimens, scoring systems and future directions are changing the approach.
You can quote PMID: 35441256 
